# Direct live imaging of cell–cell protein transfer by transient outer membrane fusion in *Myxococcus xanthus*

**Adrien Ducret[1†a], Betty Fleuchot[1], Ptissam Bergam[2†b], Tâm Mignot[1]***

[1]Laboratoire de Chimie Bactérienne, Aix Marseille University-CNRS UMR7283, Marseille, France; [2]Plateforme de Microscopie, Institut de Microbiologie de la Méditerranée, Marseille, France

**Abstract** In bacteria, multicellular behaviors are regulated by cell–cell signaling through the exchange of both diffusible and contact-dependent signals. In a multicellular context, *Myxococcus* cells can share outer membrane (OM) materials by an unknown mechanism involving the *traAB* genes and gliding motility. Using live imaging, we show for the first time that transient contacts between two cells are sufficient to transfer OM materials, proteins and lipids, at high efficiency. Transfer was associated with the formation of dynamic OM tubes, strongly suggesting that transfer results from the local fusion of the OMs of two transferring cells. Last, large amounts of OM materials were released in slime trails deposited by gliding cells. Since cells tend to follow trails laid by other cells, slime-driven OM material exchange may be an important stigmergic regulation of *Myxococcus* social behaviors.

**\*For correspondence:** tmignot@imm.cnrs.fr

[†]**Present address:** [a]Department of Biology, Indiana University, Bloomington, United States; [b]Institut Curie CNRS UMR144, Paris, France

**Competing interests:** The authors declare that no competing interests exist.

**Reviewing editor**: Peter Greenberg, University of Washington, United States

## Introduction

*Myxococcus xanthus*, a gram negative deltaproteobacterium, displays complex multicellular behaviors in response to environmental cues such as the presence of prey bacteria or starvation (*Zhang et al., 2012*). In particular, starvation triggers a developmental program where thousands of cells coordinate their motility, moving into aggregation centers to build multicellular fruiting bodies where the cells form metabolically-inert spores. This multicellular response requires an arsenal of intercellular signals, including diffusible long-range signals as well as contact-dependent signals (*Konovalova et al., 2010*; *Mauriello et al., 2009*). One intriguing cell–cell communication mechanism involves the cell-to-cell transfer of outer membrane (OM) proteins between *Myxococcus* cells. This phenomenon was originally unmasked by mixing experiments where certain motility mutants were shown to rescue other motility mutants in a process called stimulation (*Nudleman et al., 2005*). Stimulatable mutants all carried mutations in genes encoding predicted OM proteins (termed *cgl* or *tgl*). Experiments with the Tgl and the CglB OM lipoproteins suggested that stimulation is transient and does not involve the exchange of genetic material, but results from the physical transfer of Tgl/Cgl proteins from donor Tgl+/Cgl+ cells to recipient Tgl−/Cgl− cells (*Nudleman et al., 2005*).

Remarkably, OM protein exchange is not restricted to motility proteins and virtually any OM protein and even lipid can be exchanged between cells (*Wei et al., 2011*; *Pathak et al., 2012*). Gliding (A−) motility has been shown to be important for transfer, but rather indirectly by promoting the formation of dense regions of aligned cells and favoring intimate cell–cell contacts (*Nudleman et al., 2005*; *Pathak et al., 2012*). The transfer process itself depends on two specific proteins, TraA and TraB (*Pathak et al., 2012*). TraA is a protein with hallmarks of yeast floculins, a class of cell surface adhesins that mediate cell–cell interactions leading to flocculation (*Smukalla et al., 2008*) and TraB is a secreted protein of unknown function with a possible peptidoglycan-binding domain. TraA and TraB must be expressed both by donor and recipient cells for transfer to occur. Consequently, Wall and colleagues

**eLife digest** Bacteria studied in the laboratory are, in general, readily amenable to culture, and they easily form colonies when grown on agar plates. In the wild, however, many bacteria exhibit a range of more complex behaviors, including the growth of super-organisms that contain many cells.

The bacterium *Myxococcus xanthus* can exist either as single cells or as a super-organism. Each cell has an inner and outer plasma membrane separated by a periplasmic space. Previous work has found that individual cells communicate with each other by exchanging the contents of their outer membranes, and that these swaps can govern multicellular behavior.

Membrane exchange is known to depend on both donor and recipient cells having the proteins TraA and TraB. TraA proteins are similar to the adhesion factors that hold cells together, and they are found in many species: this suggests that TraA therefore might help the outer membranes of cells to fuse so that they can swap materials. The role of TraB is not known at present.

To investigate membrane exchange more closely, Ducret et al. measured the transfer of fluorescent proteins from the periplasm and the inner and outer membranes of the donor cell to the recipient cell, as well as the transfer of fluorescent lipids from the donor's outer membrane. Both proteins and lipids from the outer membrane were transferred rapidly (within minutes); although a small amount of protein transfer from the periplasmic space was observed after 36 hr, there was no transfer from the inner membrane. As in previous studies, exchange depended on the presence of TraA.

Ducret et al. observed that contact between two cells was sufficient to stimulate transfer of proteins and lipids from the outer membrane. But not all contacts led to a transfer. Importantly, when cells that had swapped fluorescent membrane components moved apart, they appeared to remain connected by tubular structures, suggesting that an inter-membrane junction must form to allow proteins and lipids to be transferred between the cells. This junction is referred to as an outer-membrane synapse.

Ducret et al. also noted another phenomenon: cells shed pieces of membrane as they moved across surfaces or separated after outer membrane exchange. This suggests that both synapse formation after direct cell-to-cell contact and the shedding of membrane components can help to propagate bacterial signals, enabling population-wide behavioral changes, including the formation or collapse of super-organisms.

proposed that when adjacent cells engage Tra-dependent surface interactions (i.e., homotypic interactions or interactions with other surface ligands), the OMs fuse locally and OM materials are exchanged (*Wei et al., 2011*; *Pathak et al., 2012*). However, because transfer was studied in bulk assays this hypothesis could not be tested directly. Therefore, other mechanisms remained possible, for example long-range exchange of OM vesicles or even local cell lysis. In this study, we investigated the transfer mechanism at the single cell level to gain more insights into the transfer mechanism.

## Results

### Transfer is a highly efficient OM-specific process

In a previous study, *Wei et al. (2011)* measured the transfer efficiency in agar plate mixing assays ('Materials and methods'), monitoring the appearance of fluorescent recipient cells over time with mCherry fluorescent probes ($OM_{mCherry}$ and $IM_{mCherry}$), which when fused to type II or type I signal sequences localize to the OM or the inner membrane (IM), respectively. However, no information was obtained about the increase in fluorescence intensity in the recipient cells. Thus, in a prelude to this study, we repeated the *Wei et al. (2011)* experiment and further measured fluorescence fluctuations in recipient cells. For completion and to test the transfer of soluble periplasmic proteins, we also constructed a periplasmic probe, fusing mCherry to the *Escherichia coli phoA* signal sequence ($PERI_{mCherry}$) ('Materials and methods' and *Figure 1—figure supplement 1*). Consistent with previous works and OM specific protein transfer, only $OM_{mCherry}$ was transferred significantly between cells. As observed by *Wei et al. (2011)*, transfer was highly efficient and ~80% of the total recipient cells were already labeled after 12 hr of co-incubation (*Figure 1A*). Transfer remained active for the next 36 hr because although the total number of recipient cells became stable after 24 hr, the fluorescence intensity of

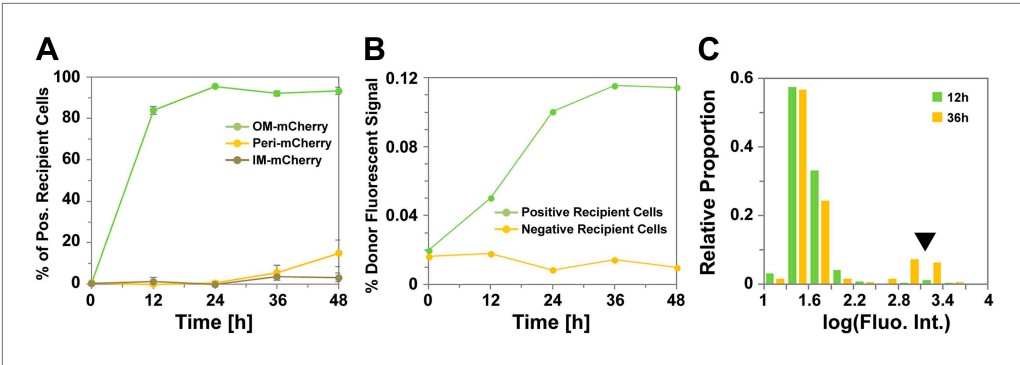

**Figure 1**. Transfer is a highly efficient OM-specific process. (**A**) Percentage of mCherry$^+$ recipient cells as a function of time. For each strain and time point, at least 3000 cells were analyzed in triplicate. Error bars = SD. (**B**) Fluorescence intensity of mCherry$^+$ recipient cells as a function of time. For each time point, the fluorescence numbers are expressed as a percentage of the mean fluorescence intensity of the donor cells population. For each time point, fluorescence intensities were measured for ~3000 cells per strain. (**C**) Distribution of fluorescence intensities measured in the positive recipient cells after 12 hr (green bars) and 36 hr (orange bars) of co-incubation. Note the logarithmic scale log(Fluorescence Intensity). The black arrow highlights the appearance of a highly-stained cell sub-population of mCherry$^+$ cells at 36 hr. For each time point, fluorescence intensities were measured for ~3000 cells per strain.

The following figure supplements are available for figure 1:

**Figure supplement 1**. Subcellular localization of indicated fluorescent probes before and after a plasmolysis treatment.

recipient cells increased regularly until it reached a plateau at 36 hr (**Figure 1B**). After 36 hr of co-incubation, 20% of the recipient cells displayed a high level of fluorescence, showing that some cells acquire exogenous OM content with very high efficiency (**Figure 1C**). A low amount of PERI$_{mCherry}$ transfer was detected after 48 hr (**Figure 1A**), suggesting that periplasmic proteins may also be exchanged but with a near background level efficiency. These findings confirm results from previous studies that transfer is a highly efficient OM-specific process.

### OM transfer can be captured at the single cell level in a live transfer assay

We next tested whether OM transfer between two cells could be captured at the single cell level. Although most of the recipient cells are stained after 12 hr, the staining is generally weak and both brilliance and the fast-bleaching of mCherry prevented single cell transfer analysis with the OM$_{mCherry}$ probe. Therefore, to maximize our chances to observe a transfer event, we constructed a new probe where super-folder GFP (sfGFP), a fast folding bright variant of GFP (**Pédelacq et al., 2006**), is fused to the type II signal sequence (OM$_{sfGFP}$). In a bulk transfer assay, OM$_{sfGFP}$ and OM$_{mCherry}$ were transferred with similar efficiencies, showing that OM$_{sfGFP}$ could be used in a single cell assay (**Figure 2—figure supplement 1A**). To this aim, *Myxococcus* donor cells expressing OM$_{sfGFP}$ were mixed with recipient cells expressing IM$_{mCherry,}$ and the emergence of dual color cells was monitored over time by time-lapse fluorescence microscopy. As observed in **Figure 2A,B** and **Figure 2—figure supplement 1D**, unlabeled recipient cells became fluorescent when they came in contact with OM$_{sfGFP}$ donor cells (**Videos 1,2**). Several lines of evidence argue that the observed fluorescence increase results from the physical transfer of OM$_{sfGFP}$: (i), Fluorescence transfer was very rapid, approximately a third of the total donor fluorescence appeared in the recipient strain after 12 min of contact (**Figure 2—figure supplement 1B**). (ii), Fluorescence initially appeared at the contact zone and subsequently diffused throughout the cell body (**Figure 2B**). Additionally, green fluorescence was enhanced at the recipient cell periphery, reflecting a membrane localization (**Figure 2—figure supplement 1C**). (iii), IM$_{mCherry}$ was not exchanged between the two cells (**Figure 2A**). (iv), Green fluorescence transfer was not detected in recipient cells that were not in contact with donor cells (**Figure 2—figure supplement 1C**), or in a negative control experiment, when they were mixed with a *traA* mutant (**Figure 2—figure supplement 1A**).

Because transfer seems highly efficient, physical transfer of OM$_{sfGFP}$ would be expected to lead to a decrease in OM$_{sfGFP}$ levels in the donor cells. While indeed a moderate decrease of sfGFP fluorescence

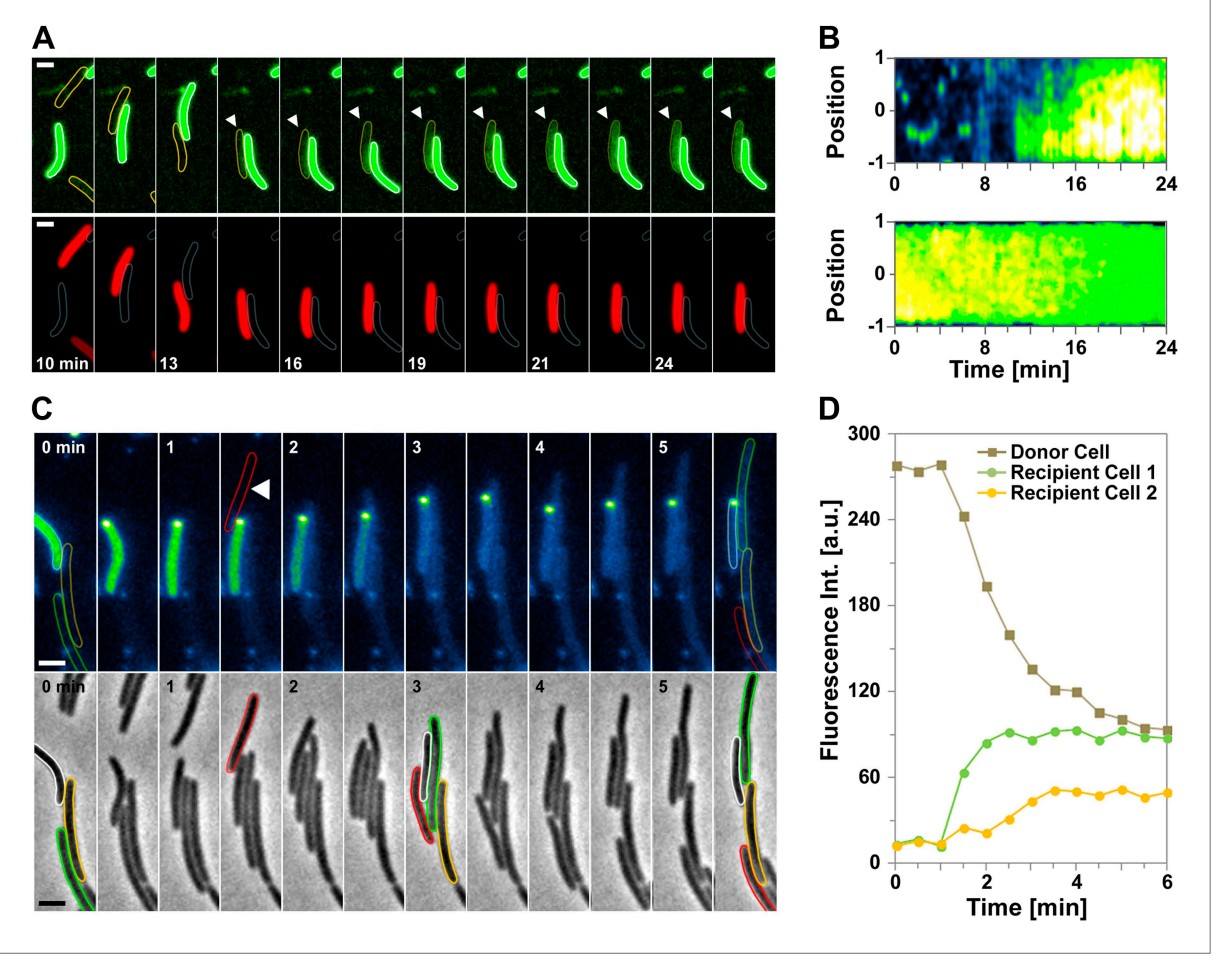

**Figure 2**. Cell-contact-dependent transfer of $OM_{sfGFP}$/DiO between single cells. (**A**) sfGFP transfer from a donor $OM_{sfGFP}^+$ (white contour in lower panel) cell to a recipient $OM_{sfGFP}^-$ $IM_{mCherry}^+$ cell (orange contour in upper panel). Scale bar=1 μm. (**B**) Kymographs of green fluorescence intensities in the positive recipient cell (top) and the donor cell (bottom) shown in (**A**). Note that in the recipient cell, green fluorescence diffuses from one half ($t_{12min}$ to $t_{16min}$) to the entire cell body. The Y-axis of each kymograph represents the relative position along the cell body, where 0 represents mid-cell and 1 or −1 the cell poles. The −1 pole is the pole closer to the bottom of the frames for each cell shown in panel (**A**). (**C**) A DiO$^+$ cell (white cell contour) transfers DiO to two unlabeled cells (orange and green contours). Fluorescence and corresponding phase contrast images are shown. Fluorescence fluctuations are shown in pseudo colors where high fluorescence levels appear yellow-green and low fluorescence levels appear blue. Note that the green cell is not immediately in contact with the DiO$^+$ cell. A cell that comes in contact with the DiO$^+$ cell but does not become labeled is shown by a red contour. Scale bar = 1 μm. (**D**) Mean DiO fluorescence intensity over time in the donor cell (gray square), the first positive recipient cell (green circle) and the second positive recipient cell (orange circle).

The following figure supplements are available for figure 2:

**Figure supplement 1**. Cell-contact-dependent transfer of OMsfGFP.

is observed in the donor cell (*Figure 2—figure supplement 1B*), the steepness of this decrease is likely compensated by the high level of newly synthesized $OM_{sfGFP}$ expressed from the strong pilin (*pilA*) promoter. To circumvent this limitation, we made use of the observation that lipids are also exchanged during transfer (*Pathak et al., 2012*) and tested the transfer of DiO, a small $C_{18}$ backbone hydrophobic lipid dye that intercalates in lipid membranes. DiO-labelled cells contain a finite amount of DiO and given that it is highly diffusible, its dilution upon transfer should be obvious. Importantly, DiO-transfer is Tra-dependent (*Pathak et al., 2012*) and thus its exchange between cells would also reflect the transfer dynamics. In a cell mixing experiment, DiO-stained cells were observed to transfer DiO to unlabeled cells upon physical contact (*Figure 2C,D*; *Video 3*). In the example shown in *Figure 2C,D*, DiO-transfer is also observed to a third cell that is not immediately in contact with the DiO-donor cell

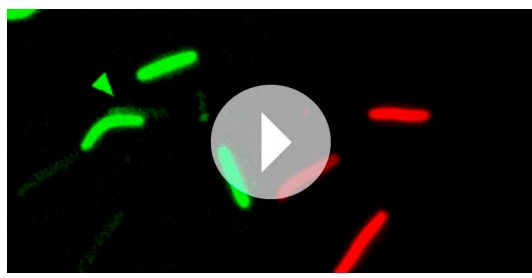

**Video 1**. Live observations of cell–contact dependent transfer of OM_sfGFP between single cells. Corresponding green fluorescence and red fluorescence are shown. For details see **Figure 2**. Pictures were taken every 30 s.

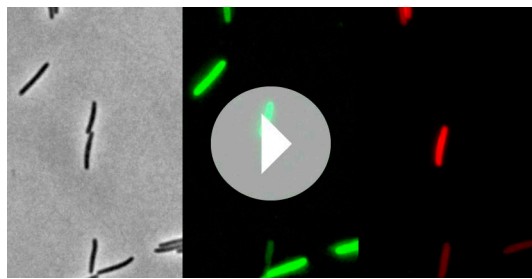

**Video 2**. Live observations of cell–contact dependent transfer of OM_sfGFP between single cells. Corresponding phase contrast, green fluorescence and red fluorescence are shown. Pictures were taken every 30 s.

but is adjacent to the first transferred cell. Remarkably, the fluorescence of the DiO-labeled cell decreased very rapidly, concomitant with the gradual increase of fluorescence in the adjacent unlabeled cells as if all three cells were connected like communicating vessels (**Figure 2C,D**). Transfer must require specific contacts (i.e., collision of TraA proteins) because unlabeled cells do not systematically acquire fluorescence when they establish a direct contact with a DiO-donor (**Figure 2C,D**; red contoured cell). In total, the OM_sfGFP and the DiO-staining experiments strongly suggest that we were able to capture transfer events at the single cell level. Transfer can occur between more than two cells, potentially explaining why it is facilitated by cell–cell alignment.

## Dynamic OM extensions are formed between cells

The DiO experiment suggests that transferring cells are connected like communicating vessels, which would be explained by the formation of transfer sites where the lipid bilayers of each OM fuse locally, giving rise to a single continuous OM between connected cells. What is the evidence for such connections? While imaging OM_sfGFP expressing cells or DiO stained cells, we frequently observed tubular structures that appeared when two connected cells moved apart (**Figure 3A**, **Video 4**). These tubes were exclusively derived from the OM because they were only stained by sfGFP when observed in two-color cells expressing

both OM_sfGFP and IM_mCherry (**Figure 3B** and **Figure 1—figure supplement 1B**). The tubes were also observed by Electron Microscopy (EM), appearing as flexible structures characterized by a diameter of 51.4 ± 15 nm (**Figure 3C** and **Figure 3—figure supplement 1A**). The structures observed by EM were not type-IV pili because (i), polar pili have a much thinner diameter (**Figure 3—figure supplement 1B**) and (ii), they were observed in a *pilA* mutant (**Figure 3—figure supplement 1C**). Interestingly, numerous tubes and vesicles were also observed in large amounts around the cells (**Figure 3C**), suggesting that lipid materials are also released by the cells (see below).

## Tube formation is linked to OM transfer

Motile transferring cells may fuse their OMs locally, forming an 'OM synapse'. If such synapses are not resolved when the cells physically separate due to motility, OM tubes would appear because of the tight physical connection. This would predict that tube formation is linked to the transfer mechanism. We first tested whether a tube and the cell OM are continuous. For this, we took advantage of the rapid diffusion of DiO and performed fluorescence recovery after photobleaching (FRAP) experiments targeting a tube connected to a single DiO+ cell. DiO fluorescence showed a quick recovery, implying rapid exchange between the tube and the cell DiO pool (**Figure 4A,B**). We then aimed to capture tube formation between transferring cells. Since the tubes are relatively short lived, we also used DiO staining for these experiments. In the example shown in **Figure 4C** and **Video 5**, DiO is exchanged upon contact between two cells, a tube becomes apparent when the cells move apart, strongly suggesting that tubes are formed between transferring cells. Last, if two cells linked by a tube have continuous OMs, they should exchange DiO, even if they are not immediately in contact. **Figure 4D** and **Video 6**, show a DiO-stained tube formed between a brightly fluorescent cell and a weakly fluorescent cell within a larger group of cells. Remarkably, in the cell with weak fluorescence, the level of fluorescence increased steadily as long as the tube connection was maintained, even though the two cells were not in

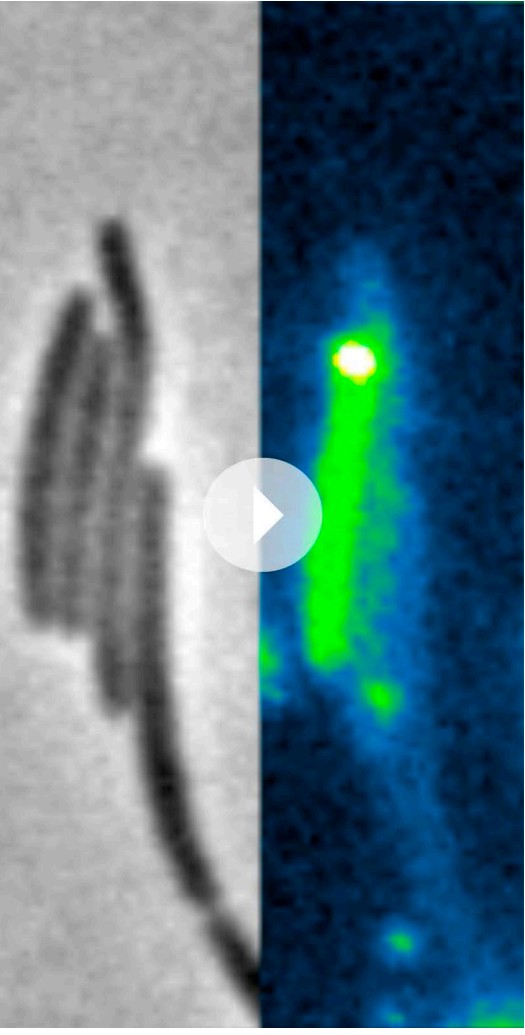

**Video 3**. Live observations of cell–contact dependent transfer of DiO between single cells. Corresponding phase contrast and green fluorescence which are displayed in pseudo colors, are shown. For details see *Figure 3A*. Pictures were taken every 30 s.

immediate contact (*Figure 4E*). When the tube was ruptured the fluorescence decreased due to photo-bleaching (*Figure 4E*). Fluorescence transfer was strictly confined to the tube-connected cells and no fluorescent fluctuations were observed in the other cells of the group (*Figure 4C,D*). Thus, tubes allow the rapid exchange of DiO and must be continuous between two connected cells.

While DiO can be exchanged through the tubes, we did not detect any significant exchange of $OM_{sfGFP}$ or $OM_{mCherry}$ through the tubes (*Figure 4—figure supplement 1*). This is probably not surprising because the tubes are narrow extensions and have a relatively short lifespan (4.2 ± 3 min). Thus, large molecules such as $OM_{sfGFP}$ or $OM_{mCherry}$ with lower diffusion rates than DiO (~fourfold, *Figure 4—figure supplement 2A,B*) may traffic slowly through the tubes. OM tubes may allow the transfer of small OM molecules, which may be relevant physiologically but they are likely the manifestation of the intimate contact established between transferring cells. The connection of cells by continuous tubes strongly argue that *Myxococcus* OM-protein transfer involves the formation of a single OM synapse between two connected cells.

## Large amounts of OM materials are deposited in slime trails during single cell motility

Where does transfer occur in the *Myxococcus* biofilm and why is it highly dependent on motility? Cell alignment in densely packed *Myxococcus* swarms promotes cell-cell transfer, likely because it favors tight interactions between cells (*Nudleman et al., 2005*; *Wei et al., 2011*; *Pathak et al., 2012*). However, Cryo-EM studies on the *Myxococcus* biofilm and our TEM and live observations of the lipid tubes also suggests that large amounts of OM materials may be released in the biofilm matrix, which may constitute a significant transfer reservoir (*Palsdottir et al., 2009*). Interestingly, when we observed gliding cells on cellulose pre-coated EM grids ('Materials and methods'), we found that cells deposit vesicular/tubular material in their wake (*Figure 5A* and *Figure 5—figure supplement 1*). This material was also observed by fluorescence microscopy and must be derived from the OM because dual labeled $OM_{sfGFP}$/$IM_{mCherry}$ cells deposited trails that were labeled with $OM_{sfGFP}$ but not with $IM_{mCherry}$ (*Figure 5B*). Gliding *Myxococcus* cells are known to deposit slime, a self-deposited sugar polymer of unknown composition that facilitates cell adhesion to the underlying substratum (*Ducret et al., 2012*). The slime polymer can be detected selectively by addition of fluorescent Concanavalin A (ConA-FITC) in a microfluidic gliding assay (*Ducret et al., 2012*). To test whether the OM materials are specifically associated with the deposited slime, we observed slime trails deposited by an $OM_{mCherry}$-expressing strain in the presence of ConA-FITC. *Figure 5B* shows that such cells deposited numerous mCherry[+] dots and tubular structures that co-localized with ConA[+] trails. EM analysis using gold-labeled ConA confirmed that the deposited OM material is embedded in a sheath of slime polymer ('Materials and methods' and *Figure 5C*). All together, these results suggest that gliding *Myxococcus* cells shed a significant amount of their OM during motility and that this material remains attached to the underlying slime polymer. Since gliding *Myxococcus* cells have long been known to follow trails left

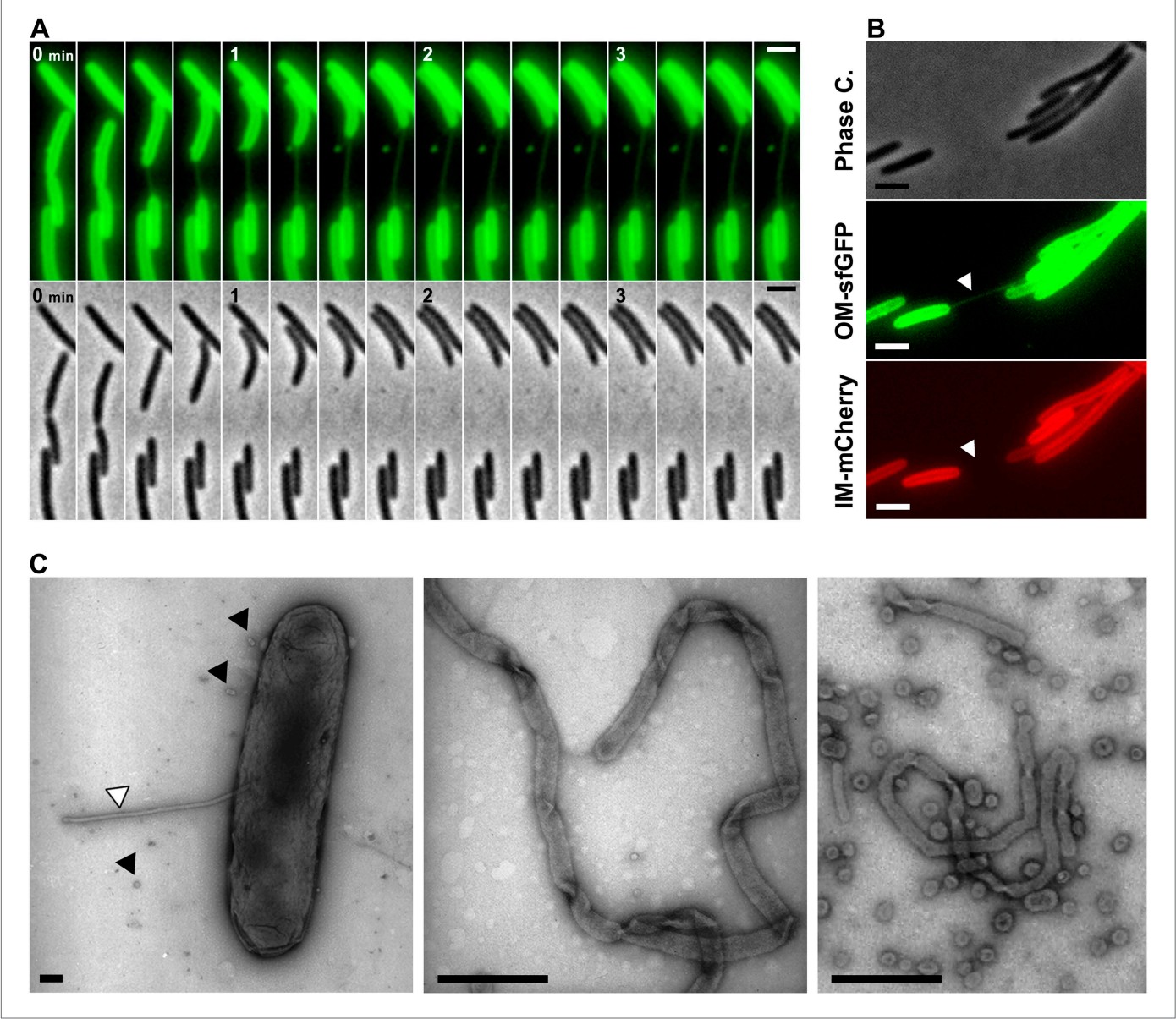

**Figure 3**. Lipid tubes are OM-derived and are observed when cells move apart. (**A**) A lipid tubes formed between two cells expressing OM$_{sfGFP}$. (**B**) Lipid tubes formed by OM$_{sfGFP}$ IM$_{mCherry}$-expressing cells (white arrow). Scale bar = 1 µm. (**C**) TEM images of lipid tubes. Tubes appear as continuous and flexible structures emerging from the cell surface (white arrow). Note the presence of vesicles in close proximity with the cell body (black arrows, left panel) or around the cells (right panel). Scale bar=250 nm.

The following figure supplements are available for figure 3:

**Figure supplement 1**. The tubular extensions are not Type-IV pili.

by other cells (*Burchard, 1982*), a tantalizing possibility is that the transfer of OM materials could also occur when cells follow slime trails, harvesting vesicles and tubes embedded in the slime. Unfortunately, we could not test this possibility directly because the amount of OM$_{sfGFP}$/OM$_{mCherry}$ labeled material remains too weak to detect a significant transfer to gliding cells by fluorescence microscopy.

## Discussion

Direct imaging of OM protein transfer between *Myxococcus* cells uncovers critical aspects of the cell biology and kinetics of transfer. Specifically, we found that the physical contact between two adjacent

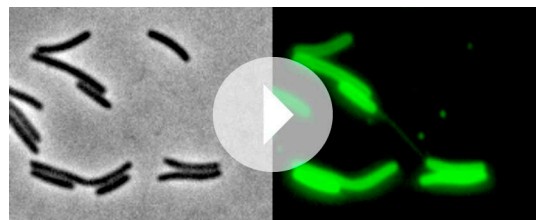

**Video 4**. Formation of OM<sub>sfGFP</sub> tubes between two cells. Corresponding phase contrast and green fluorescence are shown. Pictures were taken every 30 s.

cells is sufficient to promote transfer of OM proteins and lipids at high efficiency. This explains the results from bulk transfer experiments (from previous works and reported herein) suggesting that transfer is a remarkably efficient process. The formation of transient OM tubes between cells is a major indication that transfer indeed occurs by OM fusion: the tubes are continuous extension of cell OM, they form between transferring cells and allow the rapid exchange of lipids. Importantly, the *Myxococcus* OM transfer system is distinct from reported bacterial nanotubes, which seem to connect the cytosolic contents of connected cells and involve a yet uncharacterized machinery (*Dubey and Ben-Yehuda, 2011*). In *Myxococcus*, the transfer process is restricted to OM proteins and lipids. Transfer only occurs in a subset of cell contact events, suggesting that it is provoked by specific contacts, for example if TraA interactions brought OMs in close apposition locally. $OM_{sfGFP}/OM_{mCherry}$ are fused to type II signal sequences and thus insert in the OM as OM lipoproteins. Since, OM lipoproteins are inserted in the inner leaflet of the OM (*Nakayama et al., 2012*), transfer must involve the fusion of both leaflets of the OM membrane, suggesting that the entire OM is exchanged locally between cells. The formation of OM synapses must therefore create continuity between the periplasmic content of transferring cells. The size of the OM synapse may be estimated from the size of the tubes (~50 nm), suggesting that the diameter of the periplasmic lumen may reach up to 20 nm (for an OM of 10–15 nm thickness [*Bayer, 1991*; *Palsdottir et al., 2009*]), providing ample space for periplasmic exchange. However, the $PERI_{mCherry}$ probe was poorly if at all exchanged and there is currently no evidence for the physiological transfer of periplasmic proteins, suggesting that OM synapses are not very permeable to periplasmic proteins.

Our results also suggest that gliding motility may facilitate transfer by promoting cell–cell alignment but also when cells follow slime trails by incorporating membrane materials embedded in the slime polymer. The shedding of large amounts of membrane materials on the underlying substrate is a common byproduct of surface motility both in eukaryotic and prokaryotic cells. For example, crawling keratinocytes also deposit their plasma membrane due to the activity of acto-myosin motors in focal adhesions (*Kirfel et al., 2003*). In *Myxococcus*, gliding (A−)motility is thought to involve OM dynamics in the form of energized deformations and/or protein movements (*Nan et al., 2010*; *Luciano et al., 2011*; *Nan et al., 2011*; *Sun et al., 2011*). Thus, OM fragments may detach to the substrate due to the interaction between the motility machinery and slime. It is possible that acquisition of the *traAB* genes allowed *Myxococcus* cells to recycle this 'waste' and co-opt it for cell–cell signaling. A tantalizing possibility would be that slime embedded vesicles contain signals that promote specific recognition, facilitate trail following and promote colony expansion in response to environmental changes.

The *Myxococcus* Tra-dependent cell–cell transfer of OM proteins is a novel mode of bacterial communication that adds to the growing repertoire of bacterial contact-dependent signaling mechanisms. Contrary to known contact dependent protein transfer systems, the type VI secretion (*Silverman et al., 2012*) or intercellular nanotubes (*Dubey and Ben-Yehuda, 2011*), the distribution of TraA suggests that Tra-dependent OM fusion is restricted to the deltaproteobacteria (*Figure 6A* and *Figure 6— figure supplement 1*). Interestingly, even in *Myxococcus xanthus* strains, the predicted extracellular N-terminal PA14 domain of TraA shows variability, while the C-terminal region, presumably involved in anchoring to the cell surface is highly conserved (*Figure 6B—figure supplement 2*). Importantly, TraA acts as both key and lock for transfer to occur (*Pathak et al., 2012*). Therefore, as already suggested, OM-transfer may have evolved to regulate interactions between cells of the same kin. *tra* mutants do not show motility or developmental defects in pure culture and thus the contribution of OM transfer to *Myxococcus* social behaviors is unclear (*Pathak et al., 2012*). Interestingly however, mixing *tra* mutants with WT cells perturbs motility and development profoundly, consistent with a role in the control population dynamics (*Pathak et al., 2012*). OM exchange by transient fusion may be more widespread than suspected, especially because it is not easily unmasked and likely does not employ a conserved molecular system. Indeed, membrane vesicles and tubes have been observed in other biofilm-forming proteobacteria (*Schooling and Beveridge, 2006*; *Schooling et al., 2009*) and could well be involved in OM exchange behaviors.

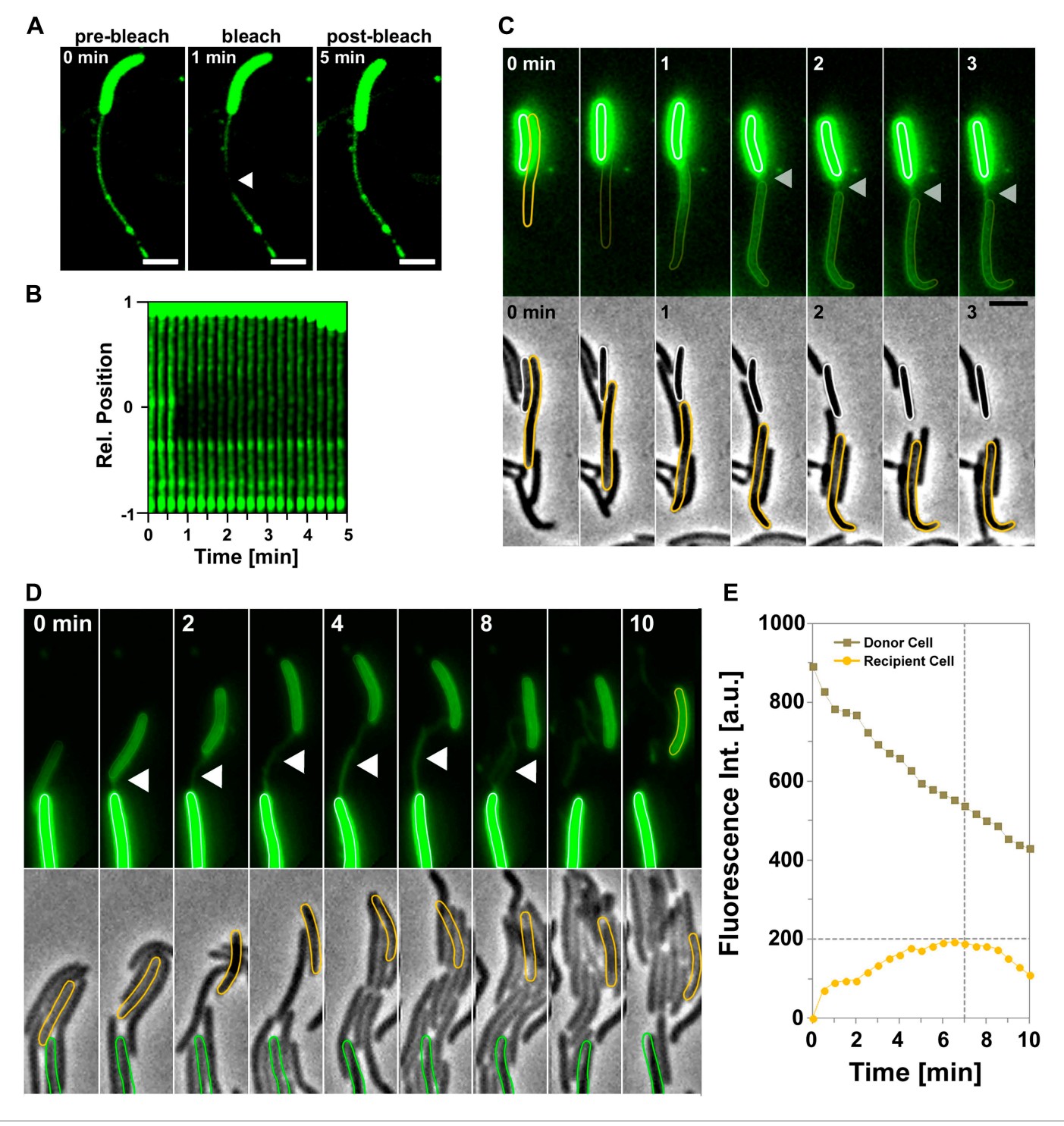

**Figure 4**. Transfer is driven by transient OM fusion between donor and recipient cells. (**A** and **B**) Fluorescence recovery after photobleaching (FRAP) experiments targeting a tube connected to a single DiO$^+$ cell. Rapid DiO exchange is observed between the tube and the cell body. The cell body is positioned at +1 in (**A**). (**C**) DiO transfer and formation of DiO$^+$ tubes between two cells. An unstained recipient cell (orange cell contour) becomes stained in contact with a DiO donor cell (white cell contour). The grey arrow points to a tube formed between the two cells. Note that transfer only occurs between the two cells although other cells are also in contact with the donor cell. Scale bar = 1 μm. (**D**) DiO is exchanged by tubes connecting two cells. Fluorescence and corresponding phase contrast images between two transferring cells (green and orange contours) are shown. Scale bar = 1 μm. (**E**) Mean DiO fluorescence intensity over time in the donor cell (gray square) and the recipient cell (orange circle). The vertical dashed line represents the

*Figure 4. Continued on next page*

*Figure 4. Continued*

time where the tube connection was ruptured. The horizontal dashed line represents the maximal value of fluorescence intensity observed in the recipient cell.

The following figure supplements are available for figure 4:

**Figure supplement 1**. The OM_sfGFP or OM_mCherry fluorescent probes are not significantly exchanged through the lipid tubes.

**Figure supplement 2**. Fluorescence Recovery After Photobleaching (FRAP) experiments targeting indicated fluorescent probes.

## Materials and methods

### Bacterial strains, plasmids and growth

Primers and plasmids used in this study are listed in *Supplementary file 1A,B*. See also *Supplementary file 1C* for strains and their mode of construction. *M. xanthus* strains were grown at 32°C in CYE rich media as previously described (*Bustamante et al., 2004*). When necessary antibiotics were added: kanamycin (Km) at 50 mg/ml, tetracycline (Tc) at 12 mg/ml, for *M. xanthus*, and Km at 50 mg/ml, or Tc at 100 mg/ml for *E. coli*. Constructs were confirmed by phenotypes, restriction analysis and DNA sequencing. Plasmids were introduced in *M. xanthus* by electroporation.

### Protein transfer experiment

For colony assays, cells were first grown in CYE, harvested and resuspended to a final concentration of $4 \times 10^9$ cfu/ml. Fluorescent donors (DZ2 PpilA–OMss–mCherry, DZ2 PpilA–IMss–mCherry, DZ2 PpilA–PERIss–mCherry or DZ2 PpilA–OMss–sfGFP) were mixed 1:1 with fluorescent recipients (DZ2 aglZ-YFP or DZ2 PpilA–IMss–mCherry). Strain mixtures were then spotted on CYE plates (1.5% agar). At various times, cells were scraped from agar plates and resuspended in TPM (10 mM Tris [pH 7.6], 8 mM MgSO$_4$, 10 mM KH$_2$PO$_4$) and spotted on agar pads to be counted directly under the micrscope. For each condition and time point, at least 3000 cells were analyzed in triplicate.

For single-cell level assays, cells were first grown in CYE, harvested, resuspended to a final concentration of $1 \times 10^7$ cfu/ml. To clearly differentiate fluorescent donor to recipient, OMss–sfGFP expressing donors was mixed 1:1 with IMss–mCherry expressing recipient. Cells were then imaged under on agar pads for up to 1 hr.

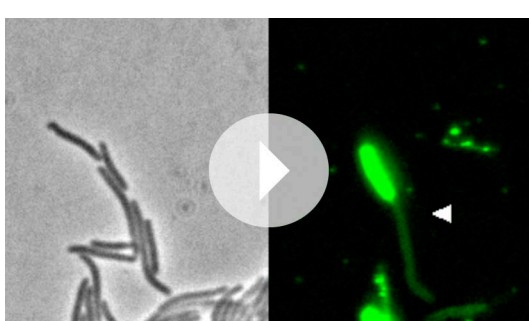

**Video 5**. Live observations of DiO transfer and formation of DiO+ tubes between two cells. For details see *Figure 3A*. Corresponding phase contrast and green fluorescence are shown. Pictures were taken every 30 s.

### Lipid dye transfer experiment

Cells were first grown in CYE, harvested, resuspended to a final concentration of $1 \times 10^7$ cfu/ml. To stain cells, 1 µl of Vybrant DiO Cell-Labeling Solution (Invitrogen, Saint Aubin, France) was added to 1 ml of cells and incubated for 30 min in the dark at 32°C under agitation. Cells were then pelleted by centrifugation, and washed four times with 1 ml TPM. Cells were then imaged on agar pads for up to 1 hr.

### Time lapse video-microscopy

Time lapse experiments were performed as previously described (*Ducret et al., 2009*). Microscopic analysis was performed using an automated and

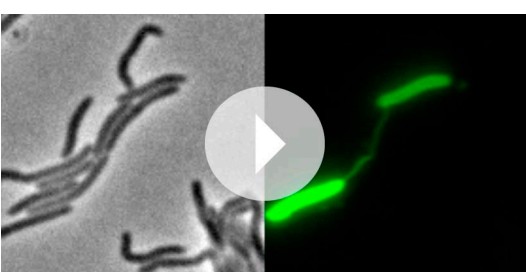

**Video 6**. Live observations of DiO transfer and formation of DiO+ tubes between two cells. For details see *Figure 3C*. Corresponding phase contrast and green fluorescence are shown. Pictures were taken every 30 s.

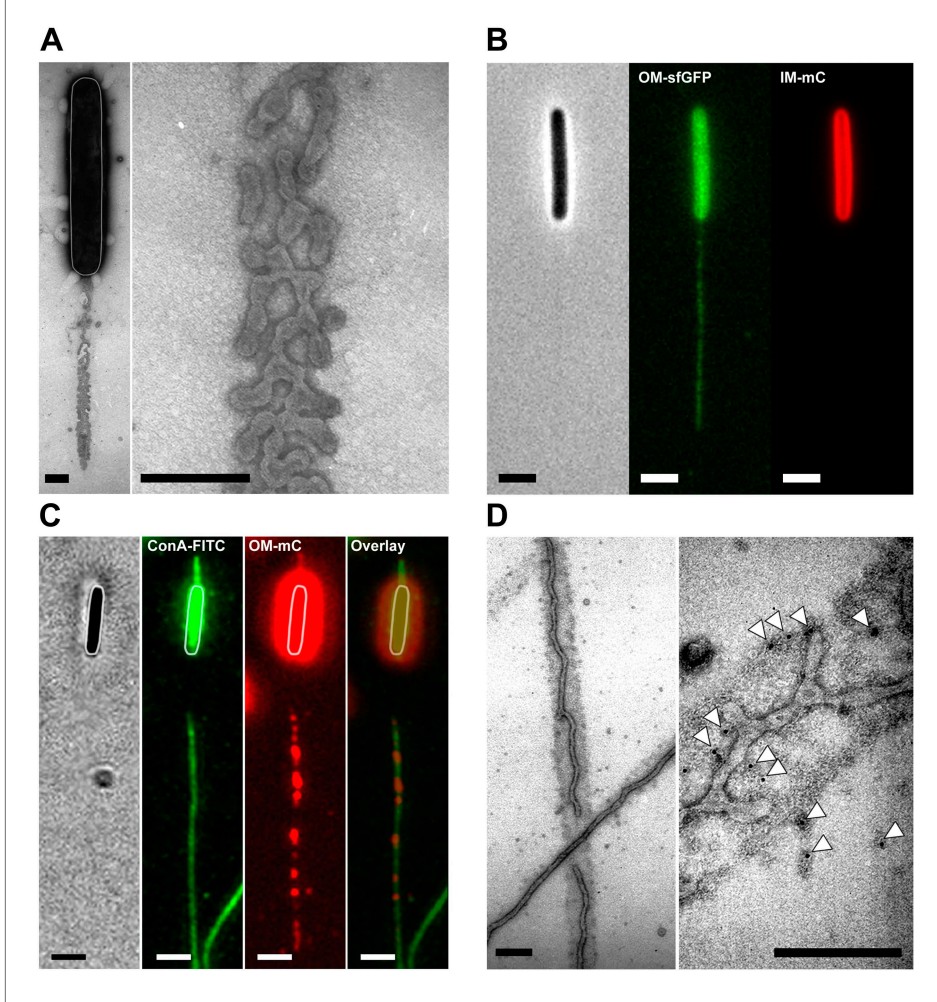

**Figure 5**. Lipid tubes and vesicles are deposited in slime trails. (**A**) TEM images of lipid tubes deposited in the wake of a moving cell (left panel). A higher magnification view of lipid tubes/vesicles is shown in the right panel. Scale bars = 250 nm. (**B**) Deposition of lipid tubes/vesicles observed by an $OM_{sfGFP}^+/IM_{mCherry}^+$ cell. The deposited material is only stained with green fluorescence implying that it is derived from the OM. Scale bar = 1 µm. (**C**) Co-localization of deposited OM materials detected using $OM_{mCherry}$ probe and slime detected using ConA-FITC. Corresponding phase contrast, red fluorescence, green fluorescence and overlay images are shown. Scale bar = 1 µm. (**D**) Lipid tubes/vesicles are embedded in the slime polymer (Black Arrow). Electron dense trails are clearly visible after ConA treatment. White arrows highlight gold particles specifically associated with biotinylated ConA and thus slime. Scale bar = 250 nm.

The following figure supplements are available for figure 5:

**Figure supplement 1**. Lipid tubes and vesicles are deposited in slime trails.

inverted epifluorescence microscope TE2000-E-PFS (Nikon, Champigny sur Marne, France). The microscope is equipped with 'The Perfect Focus System' (PFS) that automatically maintains focus so that the point of interest within a specimen is always kept in sharp focus at all times, in spite of any mechanical or thermal perturbations. Photobleaching was performed with a 488 nm laser. The bleach region of interest (ROI) was a circular region ~1 µm diameter. The ROI was uniformly bleached with a 200 ms laser exposition at 100% intensity. Images were recorded with a CoolSNAP HQ 2 (Roper Scientific, Roper Scientific SARL, France) and a 100x/1.4 DLL objective. All fluorescence images were acquired with appropriate filters with a minimal exposure time to minimize bleaching and phototoxicity effects. Cell tracking was performed automatically using a previously described macro under the METAMORPH software (Molecular devices, Evry, France) (**Ducret et al., 2009**). Typically, the images were equalized,

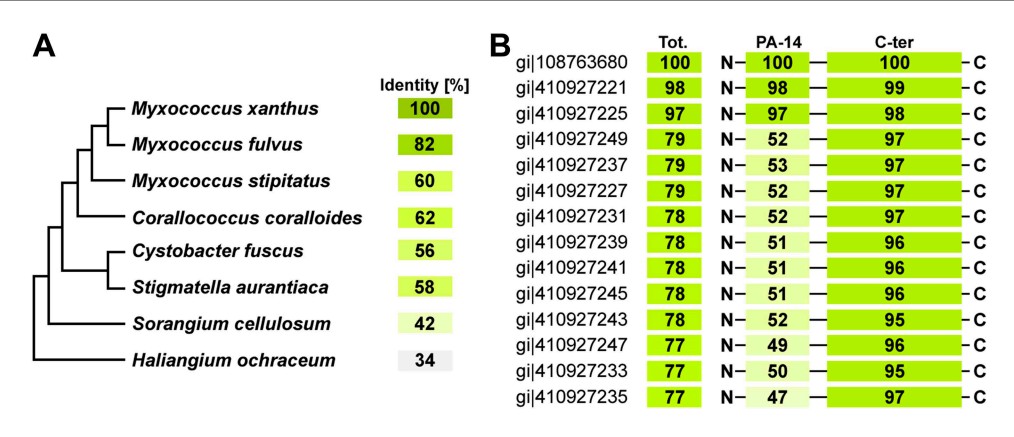

**Figure 6**. Distribution of TraA is restricted to the deltaproteobacteria. (**A**) TraA homologues in *Myxococcus xanthus* DK1622 (gi|108763680), *Myxococcus stipitatus* (gi|442324418), *Corallococcus coralloides* (gi|383459429), *Myxococcus fulvus* (gi|338532052), *Stigmatella aurantiaca* (gi|310818240), *Cystobacter fuscus* (gi|444910311), *Haliangium Ochraceum* (gi|262197466), *Sorangium Cellulosum* (gi|162451690). (**B**) The PA-14 domain is variable in *Myxococcus xanthus* strains. The conservation of the Ct domain is shown for comparison. Sequence database access numbers are shown to the left.

The following figure supplements are available for figure 6:

**Figure supplement 1**. TraA homologues in deltaproteobacteria.

**Figure supplement 2**. ClustalW alignment of TraA in *Myxococcus xanthus* strains.

straightened and overlaid under both ImageJ 1.40g (National Institute of Health, United States) and METAMORPH. Kymographs display the maximum intensity values of green or red signal along the long axis of the cell for each frame, using a 0.2 μm wide region. The Y-axis of each kymograph represents the relative position along the cell body, where 0 represents the mid-cell and 1 or −1 the cell poles. The −1 pole is always the pole closer to the bottom of the frames shown in panel.

## Diffusion rates of DiO dye and OM$_{sfGFP}$ fusion

To know the respective diffusions rates of the probes used in this study, we measured the diffusion constant of the DiO and the outer membrane probe OM$_{sfGFP}$ using fluorescence recovery after photobleaching (FRAP). As observed in *Figure 4—figure supplement 2A,B*, FRAP analysis provided a diffusion coefficient for DiO ($D_{DiO}$ = 8.1 ± 1.3 μm²/s) at least four times faster than the OM$_{sfGFP}$ probe ($D_{sfGFP}$ = 2.3 ± 0.5 μm²/s). These values are similar to diffusion constants measured for respectively outer membrane probes and outer membrane proteins in *E. coli* (*Tocanne et al., 1994*; *Chow et al., 2012*).

## TEM procedure

For TEM experiments carbon-coated copper grids were first coated with carboxymethylcellulose. Briefly, carbon-coated copper grids were covered with 30 μl of carboxymethylcellulose sodium salt (Medium viscosity, Sigma-Aldrich, Inc., St Louis, MO) diluted in ultrapure water. After 15 min of incubation at room temperature, the coating solution was removed by performing two successive washes with ultrapure water. Carbon-coated copper grids were then covered with the cell suspension previously washed and resuspended in TPM containing 100 mM of CaCl$_2$ (TPM-Ca²⁺). After 1 or 15 min incubation, unattached cells were removed by performing two successive washes with TPM. For Lectin-Gold staining procedure, carbon-coated copper grids were first covered for 30 min with 30 μl of ConcanavalinA (ConA)-Biotin conjugated (ConA-Biotin; Sigma-Aldrich, Inc.) diluted in TPM-Ca²⁺ to a final concentration of 100 μg/ml and then washed four times with TPM-Ca²⁺. The grids were then incubated for 15 min with 10 nm gold-conjugated streptavidin (Invitrogen, Saint Aubin, France) diluted in TPM-Ca²⁺ (1/500) and then washed four times with TPM-Ca²⁺. Grids were postfixed with 1% glutaraldehyde, washed once with TPM, washed four times with water, stained with 1% (wt/vol) uranyl acetate, dried, and imaged with a JEM-1011 transmission electron microscope operated at 100 kV. Cells were first observed on standard TEM

grids. As observed in *Figure 3C*, tubes appear as continuous and flexible structures emerging from the cell surface. Unattached tubes and vesicles were also observed in the vicinity of cells suggesting that this material was also released by cells in the media. On standard uncoated TEM grids *Myxococcus* cells do not glide (data not shown) precluding any observation of these structures when cells were moving. To deal with this limitation we pre-coated the TEM grids with cellulose, a linear polysaccharide composed of β(1→4) linked D-glucose units and previously known to support gliding motility of *M. xanthus* (*Ducret et al., 2013*). As observed on *Figure 5A,D* and *Figure 5—figure supplement 1A,B*, linear depositions of tubes and vesicles were observed in the wake of motile cells when cells were deposited on pre-coated TEM grids and incubated for 20 min before fixation. Linear depositions were not observed (i) when motile cells were fixed directly after deposition, (ii) with non-motile cells, and (iii) with A− cells (A⁻S⁺ strain), strongly suggesting that these depositions are specifically associated with the A-motility. Since gliding *Myxococcus* cells are known to deposit slime, a self-deposited sugar polymer that facilitates cell adhesion to the underlying substratum, we then tested if the deposited material is associated with slime. The slime polymer can be detected selectively by addition of Concanavalin A (ConA). When ConA was added to the TEM grids, electron-dense trails appeared in the wake of motile cells. The tubes and vesicles were clearly embedded in these trails. To verify that the trails are indeed labeled by ConA, we use Biotinylated ConA and colloidal gold-streptavidin. As observed on *Figure 5D*, gold particles were exclusively associated with the trail proving that vesicles and tubes are associated with the slime polymer.

## Periplasmic probe and OM/IM probe Verification—plasmolysis

To verify the proper localization of each probe, cells expressing OM$_{mCherry}$, IM$_{mCherry}$, PERI$_{mCherry}$ or OM$_{sfGFP}$/IM$_{mCherry}$ were subjected to plasmolysis (*Lewenza et al., 2008*; *Wei et al., 2011*). As predicted, OM$_{mCherry}$, OM$_{sfGFP}$ and PERI$_{mCherry}$ retained their envelope localization when IM$_{mCherry}$ probe formed fluorescent cytoplasmic aggregates (*Figure 1—figure supplement 1A,B*), indicating that following plasmolysis only the IM fusions collapse with the inner membrane. Plasmolysis was performed as previously described (*Lewenza et al., 2008*). Briefly, log phase cells were washed, resuspended in TPM buffer and then immobilized in a hybrid flow chamber (*Ducret et al., 2009*). Cells were imaged before (control) and after injection of the plasmolysis solution (0.5 M NaCl).

## Lectin staining procedure

Lectin staining was performed as previously described (*Ducret et al., 2012*). Briefly, cells were injected in a flow chamber pre-coated with Chitosan. Immediately prior to the experiments, the Concanavalin-A stock solution were diluted to a final concentration of 20 µg/ml in TPM containing 100 mM of CaCl$_2$ and 100 µg/ml of bovine *serum albumin* (BSA). The mixture was then injected into the flow chamber. After 20 min of incubation, the lectins were washed out with TPM.

## Acknowledgements

We wish to thank Emilia Mauriello, Thierry Doan, Arnaud Chastanet, Velocity Hughes and Cécile Berne for comments and discussion about the manuscript. We would like to thank Alain Bernadac for his help with TEM microscopy.

## Additional information

### Funding

| Funder | Grant reference number | Author |
| --- | --- | --- |
| HFSP young investigator grant | RGY0075/2008 | Adrien Ducret, Tâm Mignot |
| ERC starting grant | DOME 261105 | Betty Fleuchot, Tâm Mignot |

The funders had no role in study design, data collection and interpretation, or the decision to submit the work for publication.

### Author contributions

AD, BF, TM, Conception and design, Acquisition of data, Analysis and interpretation of data, Drafting or revising the article; PB, Acquisition of data, Analysis and interpretation of data

## Additional files

**Supplementary files**
• Supplementary file 1. (**A**) Strains used in this study. (**B**) Primers used in this study. (**C**) Plasmids used in this study and their mode of construction.

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
