## [Decision Letter]

Thank you for sending your work entitled “Direct Live Imaging of Cell–Cell Protein Transfer by Transient Outer Membrane Fusion in *Myxococcus xanthus*” for consideration at *eLife*. Your article has been evaluated by a Senior editor and 3 reviewers, one of whom is a member of our Board of Reviewing Editors.

Each of the reviewers has a unique perspective on which they based their comments. The Reviewing editor and the other reviewers discussed their comments before we reached this decision, and the Reviewing editor has assembled the following comments to help you prepare a revised submission.

The manuscript is an elegant and substantial contribution. The presentation made it more difficult than needed to discriminate the contributions from previous work published by Wall’s group. As written it could be viewed as a clear confirmation of the Wall findings. We believe you have made important contributions beyond the Wall papers and by revising your manuscript you can make these contributions clear. Wall predicted outer membrane tubes connecting cells and allowing transfer of material. You clearly show these tubes and characterize them (length, diameter, etc). You also provide information about the rather rapid rate of transfer from one cell to another during connection (presumably through the tubes). You also show that there is OM material in the slime trails left in the path of a cell. Although you don’t know the biological significance of this yet, it is something that we believe might be very important. It seems the work advances our understanding of signaling in *Myxococcus* and it seems that it is at least as important to those interested in OM vesicles and signaling, an emerging area on microbiology. One of the reviewers suggested that by breaking your manuscript into sections with subheadings, the special contributions of your work would become more evident.

We have highlighted a selection of the reviewers’ minor comments below, but all, including typos, require attention:

1) To recast with subheadings would likely require some reorganization because the authors move back and forth between experimental-based fact and conjecture quite easily and too often. Please be clear about what are the contributions and advances, versus confirmation and speculation.

2) Do we really know green-beard genes are extremely rare? Or have we just not looked for them much at the molecular level. Is the green beard discussion relevant? It seems quite speculative. It would be appropriate to dedicate a paragraph of the manuscript to green beards *if* the social aspect were part of the work.

3) Previous work does not appear to have strongly suggested that isolated cells cannot transfer OM material. While the papers cited (26; 18) do discuss OM transfer in the context of a biofilm, transfers like those documented here aren’t necessarily unexpected.

4) Do we still use the term “gram negative”? Hasn’t the term proteobacteria replaced this?

---

## [Author Response]

In the revised version, we have followed all the editorial recommendations to improve the clarity of the manuscript and to adapt it to a general audience. We describe the changes we made below.

The initial submission did not make it easy to discriminate previous contributions from new findings. To remedy this problem we have made substantial modifications to the structure and several parts of the manuscript. For improved clarity, we have followed suggestions to cut the text into sections and adopted a traditional “Introduction/Results/Discussion” structure. The Results section was also re-organized into paragraphs separated by subheadings. In addition, we also significantly rewrote many sections, mainly in the Introduction and Discussion sections. In the Introduction, the second paragraph now presents previous contributions from other laboratories and the proposed transfer mechanism in detail to clarify knowledge of the transfer mechanism prior to this work. Similarly, in the Discussion section, the first two paragraphs have been rewritten to discuss the extensions that our results provide to the understanding of the transfer mechanism and its biological role.

We have removed any misleading statements that previous works had suggested that transfer could not occur between single cells. We also removed the discussion on green beard genes as we agree that it was speculative. All other minor problems, grammar, problems in the figure and legends, typos etc, have been fixed as well.

We did not replace “gram negative” by “proteobacteria” because gram negative is still largely used in bacteriology and to a large community it refers quite naturally to bacteria with an outer membrane.

Finally, because *eLife* requires figure supplements to be linked to main text figures, we created a new Figure 6 to show the variability of Tra homologues in the deltaproteobacteria. The Clustal alignments occupied too much space for a main text figure and they are presented as supplements to Figure 6.